

# Preoperative 3D printing planning technology combined with orthopedic surgical robot-assisted minimally invasive screw fixation for the treatment of pelvic fractures: a retrospective study

YuLong Jing[1], LiMing Chang[2], Bo Cong[1], JianHang Wang[1], MingQi Chen[1], ZhiFeng Tang[1], JingJie Luan[1], ZiYin Han[1], YangDe Liu[1] and Tao Sun[1]

[1] Department of Traumatic Orthopedics, Yantaishan Hospital, Yantai, China
[2] Yantai Key Laboratory for Repair and Reconstruction of Bone & Joint, Yantai, China

## ABSTRACT

**Objective**. To explore the advantages and effectiveness of preoperative 3D printing planning technology combined with orthopedic surgical robot-assisted screw placement in the minimally invasive treatment of pelvic fractures compared to orthopedic surgical robot-assisted screw placement alone.

**Methods**. A retrospective analysis of the clinical data of 29 patients with unstable pelvic fractures treated with orthopedic surgical robot-assisted percutaneous screw fixation from July 2021 to August 2023 was conducted. Among them, 13 patients who underwent preoperative 3D printing technology for screw planning were assigned to the experimental group, and the remaining 16 patients were assigned to the control group. All patients underwent screw fixation alone or combined with other fixation methods for fracture fixation. The application of preoperative 3D printing planning in orthopedic surgical robot operations was described. The intraoperative screw drawing time, invasive operation time, number of fluoroscopies during invasive operation, postoperative evaluation of screw accuracy, fracture healing, complications, and functional outcomes were recorded and compared between the two groups.

**Results**. All patients successfully underwent surgery, with one patient in the control group experiencing numbness in the sciatic nerve innervation area. All patients were followed up for 4–15 months, with an average of 8 months, and all fractures achieved healing. The experimental group had a total of 26 screws inserted, while the control group had 30 screws. In the experimental group, the intraoperative screw drawing time was 3.0 (3.0, 3.37) min, significantly shorter than 4.0 (3.6, 4.0) min in the control group ($P < 0.05$). The proportion of screws not penetrating the bone postoperatively was 88.5% in the experimental group, significantly higher than 63.3% in the control group ($P < 0.05$). In the experimental group, the postoperative screw position, compared to the planned screw position, had an average position deviation of $3.05 \pm 0.673$ mm and an average spatial angle deviation of $2.22 \pm 0.605°$. At the last follow-up, the Majeed score was used to assess function, with the experimental group having an excellent and good rate of 84.6%, slightly higher than 75.0% in the control group, but the difference was not statistically significant ($P > 0.05$).

Corresponding author
Tao Sun, ytsyykyk@bzmc.edu.cn

**Conclusion**. In the treatment of pelvic fractures using screw fixation, preoperative 3D printing technology planning combined with orthopedic surgical robots, compared to orthopedic surgical robot-assisted screw placement alone, can significantly reduce intraoperative screw drawing time, decrease drawing difficulty, enhance screw placement accuracy, and does not increase invasive operation time or the number of fluoroscopies. This approach makes the surgery safer and is a method worth applying.

## INTRODUCTION

Pelvic fractures are frequently encountered clinical injuries, commonly resulting from high-energy trauma such as motor vehicle accidents and falls. These injuries are often accompanied by multiple site injuries and have a high mortality rate. The majority of pelvic fracture patients who have undergone initial resuscitation require further surgical intervention. Unstable pelvic fractures compromise the stability of the pelvic ring structure, necessitating structural reconstruction. Adequate reduction of pelvic fractures and stable reconstruction of the pelvic ring are crucial for ensuring optimal functional recovery in patients (*Barrett et al., 2014*; *Holstein et al., 2012*; *Rommens et al., 2015*). Various fixation methods exist for treating pelvic fractures, including screws, bone plates, and external fixators. However, there remains ongoing debate regarding the ideal fixation technique. While open reduction and internal fixation with plates have demonstrated favorable clinical outcomes, this method is associated with certain drawbacks, including prolonged operation times, extensive soft tissue dissection, and considerable blood loss. These factors contribute to a range of surgical complications, such as neurovascular injury and wound infection (*Grotz et al., 2005*; *Poenaru et al., 2015*). Since *Chip Routt et al. (1993)* first described and applied the percutaneous screw fixation method in 1993, numerous scholars have found that the advantages of screw fixation are becoming increasingly evident (*Florio et al., 2020*), especially for pelvic fracture patients who do not require plate-assisted reduction. A meta-analysis (*Kim & Kim, 2020*) discovered that for posterior pelvic ring fractures, screw fixation has advantages over plate fixation in terms of surgical trauma and functional recovery. Through cadaveric studies, *Hempen et al. (2022)* demonstrated that in unstable pelvic fractures caused by lateral compression forces, screws are superior to plates in terms of fracture fixation strength. Similarly, *Metz, Bledsoe & Moed, (2018)* and *Simonian et al. (1994)* also confirmed the advantages of screws in the fixation of unstable pelvic fractures caused by anterior-posterior compression forces. However, due to the complex bony structure of the pelvis and frequent deformities and variations, it is difficult to find and determine accurate screw trajectories using routine intraoperative fluoroscopy. Unstable fixation and internal fixation failure can increase the risk of nonunion, thus affecting long-term outcomes (*Archdeacon et al., 2018*; *Mitchell, Joseph & Barnard, 2015*; *Stover, Edelstein & Matta, 2017*; *Dalstra & Huiskes, 1995*). At the same time, there are many important neurovascular structures adjacent to the pelvis, which increases the surgical

risk of screw placement. Improper placement of sacroiliac screws can lead to numerous complications, with the risk of nerve injury ranging from 0.5% to 7.7% (*Altman, Jones & Routt Jr, 1999*; *Templeman et al., 1996*).

In recent years, the rapid advancement of digital technologies has significantly transformed the field of orthopedic surgery. Surgical procedures that were once extremely challenging, if not impossible, to perform manually have become more feasible and precise. Innovations such as surgical navigation systems and enhanced intraoperative fluoroscopy technologies have not only improved the accuracy of surgical interventions but have also expanded the boundaries of what can be achieved in the operating room (*Baumann et al., 2022*; *Thakkar et al., 2017*; *Boudissa et al., 2022*; *Passias et al. Benjamin, 2021*). These advancements have opened new avenues for performing intricate procedures that would be difficult or risky with conventional techniques alone. Specifically, in the context of pelvic fractures, the integration of orthopedic surgical robot navigation technology with intraoperative fluoroscopic imaging has gained considerable momentum in recent years. This combination has been shown to enhance the precision and speed of percutaneous screw placement for the treatment of unstable pelvic fractures. Numerous studies have confirmed that robot-assisted screw placement, guided by real-time imaging, offers superior accuracy and efficiency compared to traditional methods (*Long et al., 2019*; *Liu et al., 2018*; *Wang et al., 2017*; *Liu et al., 2019*). However, the combination of 3D printing technology with robot-assisted surgery represents a truly novel leap forward in surgical innovation (*Wang et al., 2020*; *Fiorini et al., 2022*). By creating patient-specific anatomical models and preoperative planning tools, 3D printing allows for a higher degree of customization, ensuring that the surgical approach is tailored to the individual's unique anatomy. When paired with the precision of robotic systems, this approach allows for unparalleled accuracy in screw placement, reduced operative time, and minimized surgical complications (*Schuijt et al., 2021*; *Suarez-Ahedo et al., 2023*). The fusion of 3D printing and robotic assistance introduces a new frontier in orthopedic surgery, providing surgeons with advanced planning tools and real-time guidance to improve outcomes in complex procedures like pelvic fracture fixation.

Through previous research, the authors found (*Jing et al., 2022*) that orthopedic surgical robot navigation technology-assisted femoral neck internal fixation has achieved good clinical results, making surgical procedures more minimally invasive and precise. However, the intraoperative planning using this technology requires manual drawing by the operator at the workstation, which has a certain degree of subjectivity and may affect the accuracy of screw insertion assisted by this technology. At the same time, 3D printing technology-assisted orthopedic surgery, especially for the treatment of pelvic fractures, has also been gradually developed in China. Preliminary applications have found that this technology can significantly improve the precision of pelvic percutaneous screw placement, not only assisting in preoperative planning but also reducing screw placement deviation and postoperative complications (*Zhou et al., 2020*; *Yao et al., 2019*; *Giovinco et al., 2012*; *Cai et al., 2018*).

Therefore, this study combines the current technological advantages by 3D printing a 1:1 pelvic fracture model preoperatively to simulate important parameters such as the

entry point and position of the screws during surgery, and applying it to real-time drawing in the intraoperative surgical robot navigation system to assist screw placement in pelvic fracture surgery, exploring its superiority. This article retrospectively analyzes the pelvic fracture patients treated with orthopedic surgical robot navigation-assisted screw fixation in our department from July 2021 to August 2023, grouping them according to whether preoperative 3D printing planning technology was used, and comparatively analyzing the advantages and disadvantages.

## METHODS

### Ethical statement

This study was approved by the Ethics Committee of Yantaishan Hospital (No. LL-2023-090-K). All cases included in the study signed an informed consent form before surgery, and patients were informed that, depending on the intraoperative situation, it may be necessary to change the surgical method, such as converting to open reduction and plate fixation or external fixator fixation.

### Patient

A retrospective analysis was conducted on 29 patients with pelvic fractures treated with orthopedic surgical robot-assisted percutaneous screw fixation in the Second Department of Traumatic Orthopedics at Yantaishan Hospital, Yantai City. Among them, 13 patients who used preoperative 3D printing technology to plan screw placement were assigned to the experimental group (*i.e.,* 3D group), and 16 patients who did not use this technology were assigned to the control group (*i.e.,* robot group). Inclusion criteria: (1) Age greater than 18 years; (2) unstable pelvic fracture; (3) closed fracture; (4) fresh fracture less than 3 weeks; (5) preoperative imaging indicates that the fracture can be partially or entirely fixed with percutaneous screws. Exclusion criteria: (1) Preoperative consideration that the fracture cannot be fixed with screws; (2) old fracture; (3) patient disagreement.

All surgical procedures were performed by physicians in our department with the title of associate chief physician or above and who have obtained a certificate of authorization for orthopedic surgical robots. The preoperative 3D printing technology planning for screw placement was also implemented by these physicians. Follow-up was conducted by physicians in our department with intermediate titles or above and with higher seniority.

### General principles for using screw fixation to treat unstable pelvic fractures

After the patient undergoes resuscitation in the emergency room, the pelvic fracture should be temporarily fixed as early as possible using an external fixator system, pelvic binder, or supracondylar femoral traction. Once the patient's condition is stable and preoperative preparations are complete, the final fracture fixation surgery is performed. Retrograde or antegrade screws are used for anterior pelvic ring injuries, and trans-sacroiliac joint screws are used for posterior pelvic ring injuries. The number of screws is determined based on the type and location of the fracture. All screws are 6.5 mm or 7.3 mm diameter short-threaded hollow screws (AO/Synthes), and the guide wire is a 2.5 mm Kirschner

wire. All surgeries are performed under general anesthesia, with the patient lying supine on a carbon fiber radiolucent table. Different reduction methods are selected according to the fracture situation. Intraoperative imaging data is obtained from C-arm X-ray fluoroscopy. For patients with anterior ring injuries, anteroposterior, inlet, and obturator outlet view fluoroscopic images are required to assist the surgery. For patients with posterior ring injuries, anteroposterior, inlet, and outlet view fluoroscopic images are required to assist the surgery. During the operation, the need for additional fixation methods, such as retaining external fixation or anterior ring internal fixation with plates, is determined based on the stability of the screw fixation.

## Surgical steps for orthopedic surgical robot-assisted percutaneous screw placement

The Ti-robot orthopedic surgical robot (Tianji 3rd generation orthopedic surgical robot; Beijing Tinavi Medical Technology Co., Ltd., Beijing, China) consists of a robotic arm, optical tracking system, surgical planning and control workstation, and surgical instruments. During the operation, the patient lies supine on a full carbon fiber surgical table, and the fracture is closed reduced and temporarily fixed through methods such as longitudinal traction of the lower limbs and external fixation. The patient tracker is fixed on the opposite anterior superior iliac spine, the surgical tracker is connected to the robotic arm, and the surgical tracker is placed on the body surface of the fracture site. The C-arm X-ray machine is used to obtain fluoroscopic images of the fracture site at specific angles, which are transmitted to the workstation. At the workstation, the screw position is drawn on the two-dimensional image according to the fracture situation. The entry point, position, angle, and length of the screw are continuously adjusted until appropriate. The information is then transmitted to control the robotic arm, guiding the robotic arm with the guiding sleeve to move to the body surface until the position and angle of the sleeve overlap with the extension line of the drawn screw. The guide wire is inserted through the guiding sleeve, an incision of about 1.5 cm is made along the guide wire, and the guide wire is gradually drilled to an appropriate length. Fluoroscopy is used to confirm that the position and length of the guide wire are satisfactory. The hole is drilled along the guide wire, and a screw of appropriate length is inserted for fixation.

## Preoperative 3D printing planning steps

Once the general condition of the patients in the experimental group is stable, pelvic computed tomography (CT) scans are performed with a slice thickness of 1 mm, and the data is transferred in DICOM format to the Mimics system (version 21.0; Materialise, Leuven, Belgium) for 3D model reconstruction. Preoperatively, fracture reduction is simulated in the Mimics system, using a 7 mm diameter cylinder to represent the screw. By rotating different visual angles, the entry point, position, angle, length, and diameter parameters of the screw are gradually adjusted. At this time, the entry point should be as far away from areas with steeper bone to reduce sliding deviation during guide wire insertion. After repeated adjustments, the optimal screw parameters are finally determined, and the screw diameter is reduced to 2.5 mm. The data is then exported in STL format to a 3D

printer (JS-600; King Stone), and a 1:1 pelvic model with guide wires is printed for later use.

## Application of preoperative 3D printing planning to orthopedic surgical robot-assisted screw placement

For the experimental group, the screw parameters have been clarified through preoperative planning using Mimics software and 1:1 model printing. During the operation, when the surgeon draws the screw at the workstation, the preoperatively planned screw entry point, position, angle, length, and diameter parameters are incorporated into the manual screw drawing, making the intraoperative screw drawing more purposeful and targeted. After the guide wire insertion is completed, the position of the guide wire is verified and confirmed using C-arm X-ray fluoroscopy images. The remaining steps are the same as the orthopedic surgical robot-assisted screw placement operation.

## Postoperative management

Routine wound dressing changes are performed postoperatively, and pelvic X-ray and CT examinations are performed within 3 days after the operation. Within 4 weeks, bed exercises are the main focus, mainly muscle contraction exercises. From 4–8 weeks, partial weight-bearing with crutches is allowed based on follow-up results. From 8–12 weeks, full weight-bearing walking is gradually allowed according to the fracture healing status during follow-up. Follow-up is conducted once a month within the first 3 months and then once every 2–3 months thereafter.

## Evaluation indicators
### Time evaluation

Record the time from the start of screw drawing to the end of screw drawing at the orthopedic surgical robot workstation, recorded as screw drawing time. For posterior ring screws, regardless of the number, record the total time; for anterior ring screws, if the number is greater than 1, record the average. When both anterior and posterior ring screws are present, record the average of the above two. Record the time from the start of guide wire insertion to the end of incision suturing, recorded as invasive operation time. If the number of inserted screws is greater than 1, record the average.

### Fluoroscopy times evaluation

Record the number of fluoroscopies from the start of guide wire insertion to the end of incision suturing, recorded as screw fluoroscopy times. If the number of inserted screws is greater than 1, record the average.

### Screw position evaluation

Evaluate after postoperative pelvic CT scan, and record the position grade of each screw for each patient according to pedicle screw evaluation criteria of *Smith et al. (2006)*. Transfer the postoperative CT data of each patient in the experimental group into Mimics software, extract the bone three-dimensional reconstruction, and separate the screws. Import the processed data together with the preoperative 3D printing planning data into Geomagic studio 2014 software (3D Systems, Rock Hill, SC, USA). After aligning the preoperative and

postoperative models using the best-fit algorithm and evaluating the alignment results using the 3D comparison function, hide the pelvic model and record the difference between the postoperative CT screw position and the preoperative 3D printing planned screw position: record as position deviation (mm) and spatial angle deviation (°), respectively. (If the number of screws is greater than 1, record the average value)

### *Functional evaluation*

Apply the *Majeed (1989)* scoring criteria to each patient at the last follow-up for postoperative functional evaluation, and record the evaluation results.

## Statistical methods

SPSS 20.0 statistical software was used for statistical analysis. Count data were expressed as rates, and comparisons between groups were performed using the chi-square test, corrected chi-square test, or Fisher's exact probability test. Measurement data underwent normality testing. Data conforming to a normal distribution were expressed as $\bar{x} \pm$ sd, and homogeneity of variance tests were performed between groups. If the variances were equal, comparisons between groups were performed using independent sample $t$-tests. Data not conforming to a normal distribution were expressed as M (Q1, Q3), and comparisons between groups were performed using the Mann–Whitney $U$ test. $P < 0.05$ was considered statistically significant.

## RESULTS

### Patient characteristics

A total of 29 patients from both groups were included in the study, and all were followed up. The median follow-up time was 8 months (range: 4 months to 15 months). In the experimental group of 13 people, there were six males and seven females, with an average age of 47.85 years. According to the Tile classification of pelvic fractures, there were 10 cases of Tile type B and three cases of Tile type C. Among the causes of injury, all 13 cases were high-energy injuries, with car accidents (10 cases) being the main cause, followed by fall injuries in two cases and heavy object crush injuries in 1 case.

The control group consisted of 16 patients, including nine males and seven females, with an average age of 47.63 years. According to the Tile classification, 14 cases of Tile type B and two cases of Tile type C were included. Similar to the experimental group, except for one case of low-energy injury due to a fall, the other 15 patients in the control group all suffered from high-energy injuries. Nine patients had fractures caused by car accidents, followed by fall injuries in three cases, heavy object crush injuries in two cases, and compression injuries in one case. The time from injury to surgery ranged from 3 days to 16 days (average 8.1 days) in the experimental group, compared to 3 days to 13 days (average 6.9 days) in the control group. Regarding multiple system injuries, in the experimental group, nine patients had concurrent chest injuries, three had craniocerebral injuries, two had maxillofacial injuries, and one had urinary system injuries. In the control group, seven patients had concurrent chest injuries, four had craniocerebral injuries, three had maxillofacial injuries, and one had urinary system injuries (Table 1).

**Table 1  Comparison of general information between the two groups.**

| Group | Cases | Age (years) | Sex (Male/Female) | Fracture type (TileB/C) | Time from injury to surgery (days) |
|---|---|---|---|---|---|
| Experimental group | 13 | $47.85 \pm 16.74$ | 6/7 | 10/3 | $8.08 \pm 4.27$ |
| Control group | 16 | $47.63 \pm 13.79$ | 9/7 | 14/2 | $6.88 \pm 2.73$ |
| Statistic | | $t = 0.039$ | $\chi^2 = 0.293$ | $\chi^2 = 0.562$ | $t = 0.920$ |
| $P$ value | | 0.969 | 0.715 | 0.632 | 0.366 |

**Table 2  Comparison of data between groups.**

| | Experimental group | Control group | Statistic | $P$ value |
|---|---|---|---|---|
| Intraoperative screw drawing time t1 (min), M(Q1,Q3) | 3.0 (3.0,3.37) | 4.0 (3.6,4.0) | $Z = -2.177$ | 0.045 |
| Invasive operation time t2 (min), M(Q1,Q3) | 20.0 (18.2,21.7) | 20.0 (19.2,21.5) | $Z = -0.956$ | 0.351 |
| Number of fluoroscopies during invasive operation n (times), M(Q1,Q3) | 9.0 (8.2,10.0) | 9.2 (9.0,10.0) | $Z = -0.954$ | 0.374 |
| Postoperative Majeed functional score (excellent and good/fair and poor) | 11/2 | 12/4 | $\chi^2 = 0.404$ | 0.663 |

## Comparison of intraoperative time and fluoroscopy times

The intraoperative invasive operation time ranged from 18 min to 22.5 min in the experimental group, with a median time of 20 min, while it ranged from 18 min to 26 min in the control group, with a median time of 20 min. The median screw drawing time during surgery was 3 min in the experimental group compared to 4 min in the control group ($P < 0.05$). The median number of screw fluoroscopies during invasive operation was nine times (range: 8–12 times) in the experimental group, compared to 9.25 times (range: 8–11 times) in the control group (Table 2).

## Postoperative evaluation of screw position

In terms of the number of screws used for pelvic fracture fixation, the experimental group had 26 screws, including 10 anterior ring screws (38.5%), while the control group had 30 screws for pelvic fracture fixation, including 15 anterior ring screws (50.0%), with no statistically significant difference between the groups. The postoperative CT evaluation of screw position was based on the Smith pedicle screw position evaluation criteria, with "excellent" representing screws completely within the bone without penetrating the cortex, "good" representing screws penetrating the cortex by less than 2 mm, "fair" representing screws penetrating the cortex by 2–4 mm, and "poor" representing screws penetrating the cortex by more than 4 mm. Among the 26 screws placed in the experimental group, 23 (88.5%) were rated as excellent, two (7.7%) as good, and 25 (96.2%) as excellent or good. In the control group, out of 30 screws, 19 (63.3%) were rated as excellent, eight (26.7%) as a good, and 27 (90.0%) as excellent or good. The rate of screws completely within the bone without cortical breakthrough was higher in the experimental group than in the control group ($P < 0.05$), and the excellent and good rate was also higher in the

**Table 3  Comparison of screw distribution, position grade, and deviation.**

|  | Experimental group | Control group | Statistic | P value |
|---|---|---|---|---|
| Total number of screws | 26 | 30 |  |  |
| Screw distribution (anterior/posterior ring) | 10/16 | 15/15 | $\chi^2 = 0.750$ | 0.386 |
| Screw position (excellent/good, fair, and poor) | 23/3 | 19/11 | $\chi^2 = 4.691$ | 0.030 |
| Postoperative *vs.* preoperative screw position deviation (mm) | $3.05 \pm 0.673$ |  |  |  |
| Postoperative *vs.* preoperative screw spatial angle deviation (°) | $2.22 \pm 0.605$ |  |  |  |

experimental group than in the control group, but there was no statistically significant difference between the groups. Postoperative screw analysis compared to preoperative planning in the experimental group: the average position deviation was 3.05 mm, and the average spatial angle deviation was 2.22° (Table 3) (typical case shown in Fig. 1).

### Final follow-up functional evaluation

At the last follow-up, pelvic function was evaluated according to the Majeed scoring criteria. In the experimental group, 11/13 (84.6%) patients had excellent or good function, while in the control group, 12/16 (75.0%) patients had excellent or good function, with no statistically significant difference.

### Postoperative complications

None of the patients in the experimental group experienced neurovascular complications after surgery. In the control group, one patient experienced numbness in the sciatic nerve innervation area after surgery, which resolved after 2 months. All patients achieved fracture healing.

## DISCUSSION

The minimally invasive treatment of pelvic fractures with screw fixation has undergone a transformation from manual screw placement to technologically-assisted screw placement. As surgical robots have gradually been applied in orthopedic surgery, their clinical benefits have always been challenging (*Lang et al., 2011*; *Gras et al., 2010*). The unique and complex anatomical structure of the pelvis makes it difficult to accurately obtain the screw insertion trajectory through intraoperative imaging, requiring repeated adjustments to prevent damage to important structures such as the L5 and S1 nerve roots, cauda equina, femoral vessels, and obturator vessels during the insertion process (*Zhao et al., 2018*; *Mirkovic et al., 1991*). Similarly, the position of screw insertion directly affects the fracture fixation outcome (*Mi et al., 2021*; *Pastor et al., 2019*). The Tianji orthopedic surgical robot system independently developed by Beijing TINAVI Medical Technologies Co., Ltd. in China, transfers the two-dimensional images of the reduced pelvic fracture during surgery to the workstation, where the surgeon draws the screw position in real-time, and then inserts the

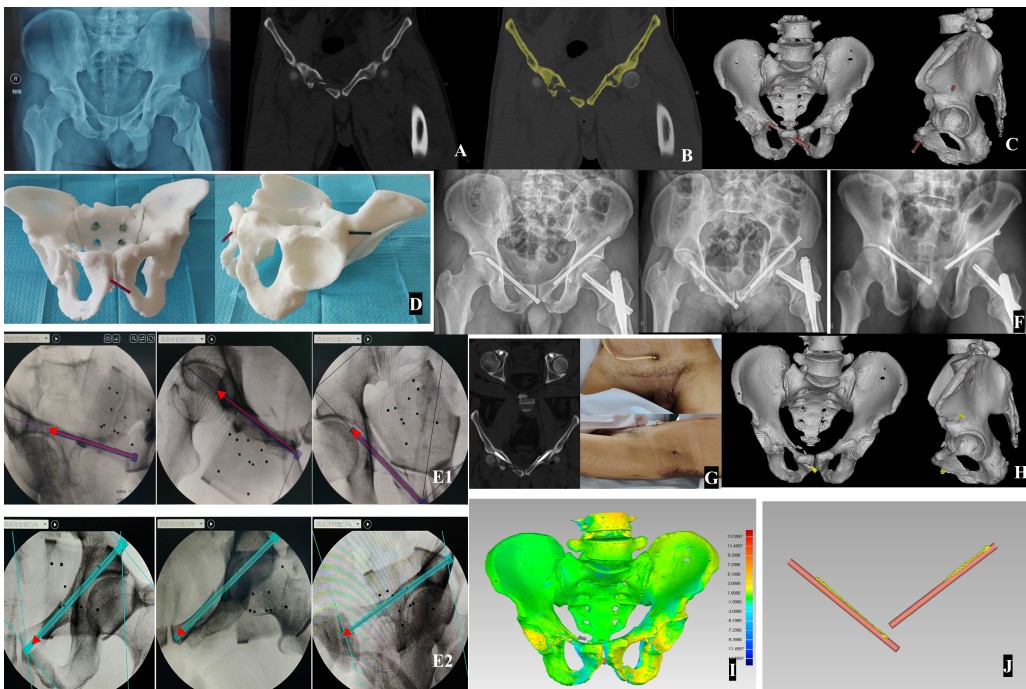

**Figure 1** **A 43-year-old male with Tile B pelvic fracture treated with screw fixation in experimental group.** (A) Preoperative X-ray and CT scan; (B) CT data were imported into Mimics software for image threshold segmentation and bone extraction; (C) pelvic three-dimensional reconstruction and simulation of reduction and screw placement (represented as brown cylinders); (D) the model was printed with guide wires, and the extension line served as an intraoperative reference; (E1-E2) intraoperative drawing of screws (dark or light blue cylinders) at the robot workstation, with the screws directions are indicated by the red arrows; (F) postoperative X-ray showed satisfactory reduction of fracture and screw position; (G) postoperative CT indicated that the screws were located within the bone without penetrating the cortex, along with the incision photo; (H) postoperative CT data were imported into Mimics software, the bone was extracted and three-dimensionally reconstructed, and then screws (in yellow color) were separated;(I) pre- and postoperative data were jointly imported into Geomagic studio 2014 software, and the the best-fit algorithm was used to align the pre- and postoperative models, and the 3D comparison function was employed to evaluate the alignment results; (J) by hiding the pelvic model, the spatial difference between the actual screw and the simulated screw was measured.

screw through the guidance of the robotic arm (*Zhu, Zheng & Zhang, 2022*). *Wang et al. (2017)* applied the orthopedic surgical robot to the screw treatment of pelvic fractures and found that compared with manual operation, the surgical robot-assisted screw insertion reduced the number of guide wire fluoroscopies and fluoroscopy time, and improved the accuracy of screw insertion without increasing the invasive operation time, which also confirmed the advantages of the surgical robot in screw insertion. At the same time, the precision and feasibility of using orthopedic surgical robots to assist screw fixation in the treatment of anterior and posterior pelvic ring fractures have also been confirmed (*Long et al., 2019*; *Liu et al., 2018*; *Liu et al., 2019*). The authors found that when using surgical robots to assist screw placement in the treatment of pelvic fractures, drawing the screw

position at the workstation during surgery needs to be completed in two-dimensional C-arm fluoroscopic images, which places high demands on the surgeon's surgical experience and is suitable for surgeons with extremely high experience in manual screw placement. Moreover, once the screw position is determined, it directly determines the subsequent operating position of the robotic arm. Some scholars have reported that incorrect drawing during surgery leads to screw penetration of the bone, requiring reoperation or changes in surgical methods (*Liu et al., 2019*). Other scholars have pointed out that repeated screw placements can damage the bony screw insertion trajectory and reduce the fixation strength (*Feng et al., 2016*). *Gardner et al. (2010)* found that small deviations in screw angle and entry point can directly lead to screw penetration of the bone, increasing the incidence of complications. *Zhou et al. (2022)* found in practice that when using antegrade pubic ramus screws to fix anterior pelvic ring fractures, the entry point is located on the more inclined bony surface above the acetabulum, which can easily lead to uncontrollable displacement during guide wire insertion, and specialized instruments must be used to reduce this problem. Moreover, intraoperative C-arm images are often affected by factors such as obesity and intestinal gas, further increasing the difficulty of intraoperative drawing. Therefore, in order to make the intraoperative drawing more targeted and the entry point selection more reasonable, the authors simulated the screw position, entry point, and other data through preoperative 3D printing to formulate a more complete and precise screw placement plan. This study found that compared with using surgical robots alone to assist screw placement, applying this technology preoperatively to simulate and formulate screw parameters significantly shortened the time for drawing screws at the surgical robot workstation during surgery ($P < 0.05$) and reduced the difficulty of intraoperative screw drawing, without increasing the invasive operation time and the number of fluoroscopies.

3D printing technology, through recent years of research and development, has been widely applied in the medical field, especially in orthopedics, where it can assist in completing personalized surgical procedures (*Zhang et al., 2021*; *Giovinco et al., 2012*; *Tack et al., 2016*). By importing personalized CT data into relevant software, not only can bone fragments be arbitrarily segmented and simulated for reduction, but the corresponding internal fixation positions and parameters can also be designed for the surgeon's reference (*You et al., 2022*). *Cai et al. (2018)* applied preoperative 3D printing to plan screw parameters and assist screw insertion in the treatment of pelvic fractures, which reduced the number of fluoroscopies and shortened the operation time compared with traditional methods. In recent years, some scholars (*Zhou et al., 2020*; *Yao et al., 2019*; *Zhou et al., 2022*) have also used preoperative 3D printing to plan screws and create personalized guide plates to assist screw placement, increasing the safety of screw placement and reducing surgical difficulty. However, how to accurately fix the guide plate during surgery remains a problem that needs to be solved. Whether it is achieved by increasing the surgical incision to expose a large area of bone or by inserting rigid fixation materials connected to the bone, such as external fixator pins, to match the guide plate base, it increases surgical trauma to a greater or lesser extent and may also increase intraoperative blood loss. At the same time, although the data source for preoperative 3D printing technology alone comes from personalized CT data, there is still a certain amount of data deviation compared to actual

surgical operations (*Banaszek, Starr & Lefaivre, 2019*; *Von Keudell, Tobert & Rodriguez, 2015*). In this study, we innovatively combined preoperative 3D printing planning with intraoperative orthopedic surgical robot operation. By comparing screw position data, we found that compared with the control group, the experimental group significantly increased the rate of screws completely located within the bone ($P < 0.05$) without reducing the excellent and good rate of screws. Without increasing the incision or adding extra intraoperative operations, the safety of the surgery was further enhanced, achieving the surgical goal of being minimally invasive and safe.

Through this study, we found that the combination of 3D printing technology and orthopedic surgical robot technology increased surgical safety and reduced surgical difficulty. However, by comparing the screw positions before and after surgery, we found that there was still an average screw distance deviation of 3.05 mm and an average screw spatial angle deviation of 2.22°, which may increase the surgical risk to a certain extent. While these deviations may appear minor, the clinical significance of these deviations warrants careful interpretation. An average screw distance deviation of 3.05 mm could lead to suboptimal fixation, particularly in cases where screws are positioned close to neurovascular structures or vital anatomical areas. Previous studies have highlighted that pelvic surgery is often performed near the spinal canal and nerves, making it easy to damage adjacent neural tissue if screws are inserted at incorrect angles (*Rexiti et al., 2020*). Misplaced screws can impinge on facet joints, potentially leading to adjacent joint degeneration, decreased postoperative quality of life, and lower back pain (*Jia et al., 2018*; *Levin et al., 2018*). Interestingly, despite the above deviations, the experimental group still had a significantly higher rate of screws completely located within the bone compared with the control group ($P < 0.05$). Some studies have shown that an average screw deviation of within 2° may not cause serious complications; other studies have found that surgical robot navigation-assisted screw placement may have a position deviation of approximately 1.9 mm (*Zhou et al., 2022*; *Krappinger, Lindtner & Benedikt, 2019*; *Chui et al., 2018*). The reasons for the deviations in this study are analyzed as follows: (1) The preoperative planning is applied to the intraoperative screw trajectory drawing, which is only the surgeon's impression of the screw, and the data and images cannot be interconnected, resulting in human deviation; (2) The intraoperative real-time two-dimensional images from the C-arm after traction reduction may have a certain deviation from the three-dimensional images formed by the preoperative CT data, leading to screw position deviation; (3) The preoperative 3D printing planning uses segmentable three-dimensional images, and the surgeon must convert them into two-dimensional image structures during the operation in order to draw, which also results in a certain human deviation.

Through this study, we found that this screw placement technology requires the surgeon to transition from preoperative planning to intraoperative drawing, inevitably resulting in subjective deviations. Therefore, this technology also has certain shortcomings: (1) Preoperative planning requires the transfer of personalized CT data and specific 3D planning software; (2) due to the limitations of the operation process, this technology can only be applied to pelvic fractures with small displacements or those that can achieve good reduction through preoperative traction and other operations and can maintain the

reduction effect relatively stably; (3) preoperative 3D printing planning requires increased treatment costs; (4) preoperative 3D printing planning data cannot be interconnected with the surgical robot workstation, and can only be transformed through the surgeon's impression. Based on this, we propose several reasonable suggestions for using this technology: (1) The preoperative planner should be the surgeon himself, who should have a certain amount of experience in pelvic fracture screw fixation; (2) for displaced fractures, preoperative CT examination should obtain data under conditions of relatively stable maintained reduction; (3) the surgeon must be able to skillfully convert between two-dimensional and three-dimensional images.

## CONCLUSION

The application of screw fixation in the treatment of pelvic fractures has made the surgery more minimally invasive, and more and more assistive technologies are being applied to the minimally invasive insertion of screws. The combination of preoperative 3D printing technology and orthopedic surgical robot-assisted screw placement is a feasible and innovative assistive technology. Through preoperative 3D printing planning, the intraoperative screw drawing time during orthopedic surgical robot-assisted surgery can be reduced, and the difficulty of screw drawing can be decreased. It also makes the selection of the entry point during surgery more targeted. Without increasing the intraoperative invasive operation time and the number of fluoroscopies, the perfect combination of preoperative 3D printing technology and robots can improve the precision of screw placement and achieve good fracture fixation, with the aim of reducing surgical complications.

### Funding
This work was funded by the Science and Technology Innovation Development Project of Yantai City, Yantai, Shandong, China (2022JCYJ037). The funders had no role in study design, data collection and analysis, decision to publish, or preparation of the manuscript.

### Grant Disclosures
The following grant information was disclosed by the authors:
Science and Technology Innovation Development Project of Yantai City, Yantai, Shandong, China: 2022JCYJ037.

### Competing Interests
The authors declare there are no competing interests.

### Author Contributions
- YuLong Jing conceived and designed the experiments, performed the experiments, analyzed the data, prepared figures and/or tables, authored or reviewed drafts of the article, and approved the final draft.

- LiMing Chang conceived and designed the experiments, analyzed the data, prepared figures and/or tables, and approved the final draft.
- Bo Cong conceived and designed the experiments, prepared figures and/or tables, and approved the final draft.
- JianHang Wang performed the experiments, prepared figures and/or tables, and approved the final draft.
- MingQi Chen performed the experiments, authored or reviewed drafts of the article, and approved the final draft.
- ZhiFeng Tang performed the experiments, authored or reviewed drafts of the article, and approved the final draft.
- JingJie Luan analyzed the data, authored or reviewed drafts of the article, and approved the final draft.
- ZiYin Han analyzed the data, authored or reviewed drafts of the article, and approved the final draft.
- YangDe Liu analyzed the data, authored or reviewed drafts of the article, and approved the final draft.
- Tao Sun conceived and designed the experiments, analyzed the data, authored or reviewed drafts of the article, and approved the final draft.

## Clinical Trial Ethics

The following information was supplied relating to ethical approvals (i.e., approving body and any reference numbers):

This study was approved by the Ethics Committee of Yantaishan Hospital (LL-2023-090-K)

## Data Availability

The raw measurements are available in the Supplementary File.

## Clinical Trial Registration

The following information was supplied regarding Clinical Trial registration:

NCT02340877

## Supplemental Information

Supplemental information for this article can be found online at http://dx.doi.org/10.7717/peerj.18632#supplemental-information.

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
