# Peer review of "Preoperative 3D printing planning technology combined with orthopedic surgical robot-assisted minimally invasive screw fixation for the treatment of pelvic fractures: a retrospective study"

_PeerJ, doi:10.7717/peerj.18632_

## Round 0.1 · original submission · Major Revisions

1. Clarify the meaning of "X" in the description of median follow-up time in the Patient Characteristics section.
2.Standardize the formatting of P values throughout the manuscript. If there is special meaning to the italics used for some, please indicate this.
3. Complete the missing part after "Data that fit a normal distribution are represented as" in the Statistical Methods section.
4. Correct the numbering format in the Inclusion Criteria section to match standard English conventions.
5. Remove the redundant parenthetical expression "(times)" after "screw fluoroscopy times" in the Evaluation Indicators section. Consider also removing "(minutes)" as it may be unnecessary.
6. Ensure all figures in Table 2 are kept to the same number of decimal places and correct the value of 3.37.
7. Improve the aesthetics and layout of Figure 1 to reduce blank spaces. Provide more detailed labels and descriptive legends to enhance clarity for readers unfamiliar with orthopedic surgery.
8. Rephrase lines 58-62 for better clarity.
9. Expand the discussion of prior technologies (lines 77-85) to better highlight the novelty of your combined 3D printing and robot-assisted surgical approach.
10. Explain the clinical significance of the reported 3.05 mm deviation (line 331) and 2.22° angular deviation (line 332) in the Discussion section.

Reviewer 1 ·

Basic reporting

NA

Experimental design

NA

Validity of the findings

NA

Additional comments

• Line 58:62 could be rephrased/ rewritten for clarity.
• The discussion of prior technologies (lines 77-85) could be expanded to better position the novelty of combining 3D printing with robot-assisted surgery.
• Figure 1 could include more detailed labels for easier interpretation. Adding more descriptive figure legends could improve clarity for readers unfamiliar with orthopedic surgery.
• Explaining the clinical significance of a 3.05 mm deviation (line 331) and 2.22° angular deviation (line 332) would enhance the discussion. While these deviations may seem minor, their impact on clinical outcomes needs to be better explained.

Reviewer 2 ·

Basic reporting

This paper compares two surgical approaches for treating pelvic fractures: preoperative 3D printing technology combined with an orthopedic surgical robot, and orthopedic surgical robot-assisted screw placement alone. The results demonstrate that the combined approach significantly improves outcomes in pelvic fracture treatment. The study is innovative and holds valuable clinical implications. While it meets the requirements for publication, several areas require further improvement.
(1)In the Patient Characteristics section regarding “Median follow-up time was X months (range: X months to X months)” what does the X here mean?
(2)For the expression of the statistical result value P, some of them are in italics, while others are not. If there is a special meaning, please indicate it, and if not, please standardize the format.
(3)There seems to be a part missing after “Data that fit a normal distribution are represented as” in the statistical methods section
(4)The numbering in the Inclusion criteria section is not in the English format, please correct it.
(5)The expression "recorded as screw fluoroscopy times (times)" in the Evaluation indicators section is redundant and need not be repeated in parentheses. In fact, the bracketed "minutes" above do not appear to be necessary either.
(6)All figures in Table 2 should be kept to the same number of decimal places and the value of 3.37 needs to be changed.
(7)The drawing of Figure 1 is not aesthetically pleasing enough, there are too many blank spaces, and it is recommended that it be changed.

Experimental design

None

Validity of the findings

None

Additional comments

None

---

## Round 0.2 · accepted · Accept

Since all comments have been fully addressed by authors, I think this paper can be accepted for publication.

Reviewer 2 ·

Basic reporting

The authors properly and clearly answered the questions, and revised the manuscript accordingly. This paper demonstrates an innovative approach by integrating preoperative 3D printing with orthopedic surgical robots, significantly advancing pelvic fracture treatment and improving clinical outcomes. I have no further comments.
Accept as is.

Experimental design

no comment

Validity of the findings

no comment

Additional comments

no comment